# Prevalence of Pneumocystosis in Sub-Saharan Africa and Helminth Immune Modulation

**DOI:** 10.3390/jof8010045

**Published:** 2021-12-31

**Authors:** Luis Fonte, María Ginori, Enrique J. Calderón, Yaxsier de Armas

**Affiliations:** 1Parasitology Department, Institute of Tropical Medicine “Pedro Kourí”, Havana 11400, Cuba; 2Department of Teaching, Polyclinic “Plaza de la Revolución”, Havana 11300, Cuba; maginorig@infomed.sld.cu; 3Instituto de Biomedicina de Sevilla, Hospital Universitario Virgen del Rocío/Consejo Superior de Investiga-Ciones Científicas/Universidad de Sevilla, 41013 Seville, Spain; 4Centro de Investigación Biomédica en Red de Epidemiología y Salud Pública (CIBERESP), 28029 Madrid, Spain; 5Department of Clinical Microbiology Diagnostic, Hospital Center of Institute of Tropical Medicine “Pedro Kourí”, Havana 11400, Cuba; Yaxsier@ipk.sld.cu; 6Pathology Department, Hospital Center of Institute of Tropical Medicine “Pedro Kourí”, Havana 11400, Cuba

**Keywords:** *Pneumocystis* pneumonia, pneumocystosis, HIV infection, helminth immune modulation, Sub-Saharan Africa

## Abstract

Sub-Saharan Africa is the region of the world with the highest prevalence of helminth infections. To protect themselves from the defensive mechanisms of their respective hosts, helminths modulate their immune responses. This modulation has relevant clinical and epidemiological consequences, including the inhibition of inflammatory processes that characterize infection by other microorganisms. Severe *Pneumocystis* pneumonia is characterized by an intense inflammatory reaction that can lead to death. Acquired immunodeficiency syndrome is the main predisposing factor to the development of pneumocystosis. Although the introduction of highly active antiretroviral therapy has led to a notable decline in the incidence of acquired immunodeficiency syndrome-associated complications, pneumocystosis continues to be an important global health problem. Despite the high incidence of human immunodeficiency virus infection in the sub-Saharan region, the prevalence of *Pneumocystis* pneumonia there has been lower than expected. Several factors, or combinations thereof, may contribute to this evolution. Here, we hypothesize the possible role of helminth immune modulation as an important issue at play. On the other hand, and looking ahead, we believe that the immune modulation achieved by helminths may be an important factor to consider during the design and evaluation processes of vaccines against *Pneumocystis jirovecii* to be used in Sub-Saharan Africa. The requirements of a balanced triggering of different types of immune responses for controlling the infection produced by this microorganism, as observed during experiments in animal models, support this final consideration.

*Pneumocystis* pneumonia (PcP), caused by *Pneumocystis jirovecii* (formerly *P. carinii sf. hominis*), is one of the most common fungal diseases in immunosuppressed persons [1]. Acquired immunodeficiency syndrome (AIDS) is the main predisposing disease to the development of PcP (also known as pneumocystosis) [1,2]. Other conditioning factors for the development of pneumocystosis include immunosuppressive therapy, neoplasms and, in the case of children, severe combined immunodeficiency [3,4]. Although the introduction of highly active antiretroviral therapy (ART) has led to a notable decrease in the incidence of complications associated with AIDS, PcP continues to be an important global health problem [1,2]. In recent years, pneumocystosis has shown a worldwide annual incidence greater than 19 cases per million inhabitants [5].

Most people have been exposed to infection by *P. jirovecii* by the age of four years (this organism is common in the air in almost all scenarios). Some studies confirm, using polymerase chain reaction (PCR) and serologic procedures, that more than 80% of children between the ages of two and four years have been in contact with this fungus [6,7]. Immunocompetent hosts commonly eliminate colonizing organisms, or they persist at a low burden, without clinical manifestations, while immunodeficient hosts cannot control the infection and develop PcP, in which an excessive inflammatory response leads to lung damage [8,9].

*P. jirovecii* remains impossible to cultivate in vitro, slowing progress in understanding numerous aspects of pneumocystosis, including host immune responses to the organism and the immunopathology of the disease. Current information about basic aspects of PcP has emerged from some limited clinical studies and, mainly, from experiments in rodent models (*P. murina* in mice and *P. carinii* in rats). Humans and rodents share dichotomous asymptomatic infection or interstitial pneumonia in immunocompetent or immunosuppressed hosts, respectively [9,10].

Some progress has been made in understanding how the host responds to and clears *Pneumocystis*. Taken together with the results of studies using gene-deficient mice, it is possible to conclude that *Pneumocystis* induces multiple T-cell-mediated responses, including Th1/Th17 and Th2 types. However, individually, these responses appear to be dispensable. This has suggested that the intrinsic pathway of T cells is likely to include multiple components that lead to a mixed cytokine environment with compensatory mechanisms [11].

Recently, Eléna Charpentier et al., in an excellent review on the immune response to *Pneumocystis* infections, summarized the subject in this form: “... the immune response associated with a favorable outcome of the infection may differ according to the immune status of the host. In the case of immunocompetency, a close communication B cells and TCD4 within tertiary lymphocyte structures appears critical to activate M2 macrophages without much inflammation. Conversely, in the case of immunodeficiency, a pro-inflammatory response including Th1 CD4, cytotoxic CD8, NK cells, and IFNγ release seems beneficial for M1 macrophage activation, despite the impact of inflammation on lung tissue” [9]. Although macrophages of both profiles can phagocyte *Pneumocystis*, those that harbor an alternative activation M2 profile (that is, activation through cytokine of the Th2 response) are preferentially involved in immunocompetent mice. This M2 profile, which may be related to the mechanisms of helminth immune modulation that we describe below, is consistent with the predominant Th2 profile reported in immunocompetent mice infected with *Pneumocystis* [12,13].

Before 1995, it was estimated that two-thirds of the HIV-infected people worldwide developed PcP [14]. Paradoxically, studies in some SSA countries with high HIV prevalence reported that PcP was uncommon [15,16,17]. Although the worldwide prevalence of pneumocystosis has declined since the introduction of cotrimoxazole prophylaxis and ART during the 1990s, PcP remains the most important opportunistic infection defining AIDS in the United States and Europe [1,18]. Contrary to that general tendency after the administration of the mentioned drugs, in the sub-Saharan region, morbidity and mortality in people infected by HIV are dominated by other infectious diseases, mainly tuberculosis [18]. Some factors, or a combination of them, have been investigated to explain the lesser prevalence of PcP in the sub-Saharan region: (i) differences in methodologies and limitations in resources (humans and materials) for PcP diagnosis [18,19,20], (ii) the virulence of strains of *P. jirovecii* circulating in SSA (genotyping at the superoxide dismutase locus of Isolates of *P. jirovecii* did not find geographic differences) [21], (iii) host resistance to *P. jirovecii* infection (a work shows an association between the CXCR6 genotype and progression from PcP to death among African Americans with HIV) [22], (iv) seasonal variations in the SSA region (data from human studies shows that, rather than a seasonal association, presentation with PcP appears to be highest when the average temperature is between 10 and 20 °C) [23], and (v) HIV-infected African adults have high rates of bacterial pneumonia and tuberculosis, diseases that can result in death at higher CD4+ cell counts and prevent many HIV-infected patients from reaching a stage at which they would be susceptible to PcP [18,19]. Here, we hypothesize the possible role of helminth immune modulation in the lower prevalence of PcP in SSA.

Sub-Saharan Africa is the region of the world with the highest prevalence of helminth infections and the highest concentration of poverty [24]. In individuals chronically infected with helminths, prolonged host–parasite coevolution has led to the development of defensive responses by human hosts and the achievement of complex immune modulatory means by helminths. To control helminth infections, adaptive immunity of the host usually develops type 2 immune responses, including Th2 cell development and cytokine release such as IL-4, IL-5 and IL-13 [25]. This host–helminth interaction has, at least, two additional outcomes: (i) the classical and best known down-regulation of the responses of type Th1 and type Th17 (and its related cytokines IL-12, IFN-γ, IL17, IL-23, TNF-α) by Th2 cytokines [25,26,27], and (ii) the limitation of helminths of both host type 1 and type 2 responses by enhancing FOXP3+ T regulatory cells, B regulatory cells, and M2 macrophages activities, which together cause the release of regulatory cytokines such as IL-10 and transforming growth factor (TGF-β) [27].

The helminths’ regulation of the immune responses of their hosts has relevant clinical and epidemiological consequences: increased susceptibility to some infections, decreased frequency and intensity of allergic, autoimmune and inflammatory diseases, inadequate responses to vaccines and, as is possible in the case of *P. jirovecii* infection, inhibiting the inflammatory processes that characterize infection by other microorganisms [26].

Helminths can limit the development of inflammatory responses during virus, bacteria, and protozoon infection: (i) in mice, chronic infection with *Schistosoma mansoni*, a blood fluke residing in the mesentery of the intestines, induces a mixed Th1 and Th2 response that protects against lung damage caused by infection with influenza A virus or a mouse model of respiratory syncytial virus (RSV) [28]; (ii) mice infected by *Nippostrongylus brasiliensis* showed increased susceptibility to *Mycobacterium tuberculosis*. Apparently, M2 macrophages with impaired killing capacity in a less inflammatory type 2 pulmonary environment function as a mycobacteria reservoir [29]; (iii) when *Plasmodium falciparum* infection occurs in an individual infected with helminths, the effects of pro-inflammatory cytokines (IFN-γ and TNF-α) that characterize severe forms of malaria are attenuated by the action of anti-inflammatory mediators (IL-10 and TGF-β) and, consequently, decrease the chances of developing severe inflammatory conditions, including cerebral malaria [30]; (iv) *Trichinella spiralis* infection limits inflammatory pulmonary damage induced by *influenza virus* in mice [31]; and (v) gastrointestinal parasites, hookworms in particular, create an immunoregulatory environment that promotes their own survival, but paradoxically also benefits the host by protecting against the progress of inflammatory diseases [32].

Taking into account the arguments described above, it is plausible to consider other factors, such as the inhibition of inflammatory processes by regulatory mechanisms induced by helminths, to provide an explanation for the lower prevalence of pneumocystosis in SSA. In line with the rationale of the hypothesis exposed here, it is interesting to mention that the severity of coronavirus disease 2019 (COVID-19), another respiratory disease related to the development of an intense pulmonary inflammatory reaction, is lower in SSA compared to Europe and the United States [33].

Racial differences in the incidence of PcP have been observed in Europe and the United States, with a lower incidence in HIV patients of African origin compared to HIV patients of Western origin [34]. Differences in genetic susceptibility have been suggested to partially explain the lower incidence of PcP in these patients [22]. Alternatively, in light of the above arguments, another explanation is possible: most people who migrate from Africa to Europe and the United States do so after adolescence, when they already bear the imprint of the regulation produced in their immunological memories by some infection by helminth. These people, akin to those who remain in SSA, arrive in their destination countries with some protection against the inflammatory phenomena that characterize some infectious diseases and autoimmune disorders, such as PcP and inflammatory bowel disease, respectively. Joel Weinstock, in an illustrative comment on the subject, explains that “Today, Native American reservations, which have relatively high rates of infection with parasitic worms, also have lower rates of inflammatory bowel disease. Latinos born and raised in South America rarely develop this gut disorder. If their children are born in the United States, where conditions are often more sanitary, they have a much higher risk of the disease” [35].

Recent advances in knowledge on immune responses to *P. jirovecii*, including factors that impact immunopathological mechanisms of pneumocystosis, allow a better understanding of important aspects of this fungal infection. According to this, confirmation of the possible influence of helminth immune modulation on PcP prevalence in the sub-Saharan region will be necessary. A better understanding of the epidemiological particularities of pneumocystosis in SSA, including the influence of coinfections on its clinical course, will allow a more rational use of the available diagnostic and therapeutic tools, very scarce in many countries in this impoverished region. On the other hand, and looking ahead, we believe that the immune modulation achieved by helminths may be an important factor to consider during the processes of design and evaluation of vaccines against *P. jirovecii* to be used in SSA. The requirements of a balanced triggering of different types of immune responses for controlling the infection produced by this microorganism, observed during experiments in animal models mentioned above, support this final consideration.

## Data Availability

Not applicable.

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
