# Peer review of "Prevalence of Pneumocystosis in Sub-Saharan Africa and Helminth Immune Modulation"

_jof, 2021, doi:10.3390/jof8010045_

Round 1

Reviewer 1 Report

This is an interesting and well-written opinion article on the possibility that high exposure to immune modulation by helminths in the sub-Saharan African population may be working as a protection against the development of severe Pneumocystis pneumonia, masking and/or decreasing the incidence of this infection in this population. 

However, the authors finish this article with a strong claim that is, throughout the article, poorly supported. The feasibility of using the immune modulation achieved by helminths as a tool to design a preventive or prophylactic tool for PcP in this population is insufficiently discussed. The authors themselves raise very relevant clinical and epidemiological limitations of this approach (lines 119-121 and lines 128-132) considering other infections/diseases that are also very prevalent in this population. Thus, a more detailed discussion to support a scientific investment in this idea is lacking.

Taking this into consideration, the conclusion should be revised or the introductory part improved in order to support this conclusion.

Author Response

Answers to Reviewer 1

To both Reviewers:

Thank you very much for your revisions, comments and suggestions. Please, find below the answers to your queries. As you will note, your commentaries and suggestions helped us to prepare a more accurate version of our manuscript.

To Reviewer 1:

  1. However, the authors finish this article with a strong claim that is, throughout the article, poorly supported. The feasibility of using the immune modulation achieved by helminths as a tool to design a preventive or prophylactic tool for PcP in this population is insufficiently discussed. The authors themselves raise very relevant clinical and epidemiological limitations of this approach (lines 119-121 and lines 128-132) considering other infections/diseases that are also very prevalent in this population. Thus, a more detailed discussion to support a scientific investment in this idea is lacking. Taking this into consideration, the conclusion should be revised or the introductory part improved in order to support this conclusion.

Answer:

For times, the prevalence of helminth infections in SSA has been very high. For surviving, helminths modulate the immune responses of their hosts. The modulation of immune responses by helminths is highly anti-inflammatory, to the point that allergic and autoimmune events in SSA are relatively rare. The severity of pneumocystosis is mainly due to inflammatory phenomena. In SSA, very different from what was expected due to the prevailing high incidence of human immunodeficiency virus infection there, the prevalence of pneumocystosis has been lower compared to Europe and Unite State. In our Opinion submitted to Journal of Fungi we hypothesize, from a holistic vision that also mentioned other factors, that helminth modulation could be influencing on the prevalence of pneumocystosis in SSA.

All the texts of our manuscript were used to argument our hypothesis; it is the objective of our paper.

In our manuscript summited to Journal of Fungi we did not discuss “the feasibility of using the immune modulation achieved by helminths as a tool to design a preventive or prophylactic tool for PcP in this population is insufficiently discussed …” In the last paragraph, we wrote two sentences for ending our manuscript in perspective (“looking ahead”): (i) “A better understanding of the epidemiological particularities of pneumocystosis in SSA, including the influence of coinfections on its clinical course, will allow a more rational use of the available diagnostic and therapeutic tools, very scarce in many countries in this impoverished region”; that is, that the confirmation of our hypothesis would allow a better use of the available diagnostic and therapeutic tools; and (ii) “On the other hand, and looking ahead, we believe that the immune modulation achieved by helminths may be an important factor to consider during the processes of design and evaluation of vaccines against P. jirovecii to be used in SSA”; that is, that in perspective it is necessary to take into account that helminth modulation may additionally impair the adequate responses to vaccines against the microorganism.

Taken into consideration your commentary, in the abstract of the new version of our manuscript we are substituting the phrase “However, in the future, we believe…” (line 28) with the phrase “On the other hand, and looking ahead, we believe…” (marked in yellow in the abstract of the new version), in harmony with the last paragraph of the main text, and reinforcing in perspective the idea that helminth immune modulation may additionally impair the response to pneumocystosis vaccines in development.

Thank again for your commentary, which has helped us to prepare a more accurate version of our Opinion.

Reviewer 2 Report

Association of respiratory disease and helminthic infection goes back to 1877, when William Osler described “Verminous Bronchitis  in dogs” (Osler W, Veterinarian, 1877; 5:387-397). Opinion manuscript by Fonte Luis et.al submitted to “Journal of Fungi” attempts to  extrapolate   plausible    immune modulating actions of  helminthic infections in the context of Pneumocystis pneumonia  (PcP),  giving an elaborate complexities and reciprocity of immunological basis.  This further  reinforces well established  studies  that  highlight  association between various diseases/disorders with helminthiasis.  

 This nicely written perspective in fact  provides impetus to carry out  population based studies in Sub-Saharan Africa (SSA)  as well as in  other developing countries  to elucidate  both risk and protection evaluation of helminthic infections and PcP.

I have the following queries towards the authors and these may kindly be addressed and incorporated--

Q.1—  It would be interesting to incorporate  about pathogenic interaction/s about helminths and non-HIV patients with PcP.

Q.2 – (In Line 96)  “ Iess  virulence of P.jirovecii strains  circulating in SSA”  may be explained providing some results of population genetics and or experimental  studies  conducted in this regard.

Q.3. More so, (in line 97), “SSA population may be more resistant to P.jirovecii infection” should be  substantiated.

Q.4. Although the associative role  of Schistosoma mansoni, Nippostrongylous brasiliensis  in relation to  airway inflammation  have been brought out, it would be worthwhile  to discuss about some other important helminths such as  Ancylostomiasis  , being more common in SSA .

Author Response

Answers to Referee 2

To both Reviewers:

Thank you very much for your revisions, comments and suggestions. Please, find below the answers to your queries. As you will note, your commentaries and suggestions helped us to prepare a more accurate version of our manuscript.

To Reviewer 2:

Q.1. It would be interesting to incorporate about pathogenic interaction/s about helminths and non-HIV patients with PcP.

Answer:

As far as we know, our manuscript is the first that relates the lower prevalence of pneumocystosis with the high endemicity of helminth infections in SSA, a region with a very high incidence of HIV infection. In response to your interesting suggestion, we have exhaustively reviewed the published information on the prevalence of pneumocystosis in areas with high endemicity of helminth infections and low incidence of HIV infection. As we expected, we did not find articles on this matter (it happens that most of the areas with a high endemicity of helminth infections also have high incidence of HIV infection). We consider that, even in the absence of HIV infection, the anti-inflammatory component of the helminth immune modulation would have a protective effect on the development of severe forms of PcP.

Q.2. (In Line 96) “Iess virulence of P.jirovecii strains circulating in SSA” may be explained providing some results of population genetics and or experimental  studies  conducted in this regard.

Q.3. More so, (in line 97), “SSA population may be more resistant to P.jirovecii infection” should be substantiated.

Answers to Q.2. and Q.3.:

For times, the prevalence of helminth infections in SSA has been very high. For surviving, helminths modulate the immune responses of their hosts. The modulation of immune responses by helminths is highly anti-inflammatory, to the point that allergic and autoimmune events in SSA are relatively rare. The severity of pneumocystosis is mainly due to inflammatory phenomena. In SSA, very different from what was expected due to the prevailing high incidence of human immunodeficiency virus infection there, the prevalence of pneumocystosis has been lower compared to Europe and Unite State. In our Opinion submitted to Journal of Fungi we hypothesize, from a holistic vision that also mentioned other factors, that helminth modulation could be influencing on the prevalence of pneumocystosis in SSA.

Afraid of using more words than those permitted in Journal of Fungi for an Opinion, we focused on the description of the arguments that support our hypothesis. In relation to the factors alluded to by other authors, about which the information is very scarce, we simply mentioned each one (from line 95 to line 102 in the first version). However, taken into consideration your suggestion, we are introducing additional information on those factors in the new version of our manuscript (marked in yellow in the corresponding paragraph of the new version). To document this additional information, we are adding further references (references 21, 22 y 23 in the new version, marked in yellow).

.Q.4. Although the associative role of Schistosoma mansoni, Nippostrongylous brasiliensis  in relation to  airway inflammation  have been brought out, it would be worthwhile  to discuss about some other important helminths such as  Ancylostomiasis, being more common in SSA.

Thank for this suggestion. Taken in consideration it, we are introducing additional information about other helminths, included hookworms, which are very prevalent in SSA, in the new version of our manuscript (marked in yellow in the corresponding paragraph of the new version). To document this additional information, we are adding further references (references 31 y 32 in the new version, marked in yellow).

Thank again for your commentaries and suggestions, those have helped us to prepare a more accurate version of our Opinion.